# Carbonate formation in salt dome cap rocks by microbial anaerobic oxidation of methane

K.H. Caesar[1], J.R. Kyle [ID] [2], T.W. Lyons[3], A. Tripati[4] & S.J. Loyd[1]

Major hydrocarbon accumulations occur in traps associated with salt domes. Whereas some of these hydrocarbons remain to be extracted for economic use, significant amounts have degraded in the subsurface, yielding mineral precipitates as byproducts. Salt domes of the Gulf of Mexico Basin typically exhibit extensive deposits of carbonate that form as cap rock atop salt structures. Despite previous efforts to model cap rock formation, the details of subsurface reactions (including the role of microorganisms) remain largely unknown. Here we show that cap rock mineral precipitation occurred via closed-system sulfate reduction, as indicated by new sulfur isotope data. $^{13}$C-depleted carbonate carbon isotope compositions and low clumped isotope-derived carbonate formation temperatures indicate that microbial, sulfate-dependent, anaerobic oxidation of methane (AOM) contributed to carbonate formation. These findings suggest that AOM serves as an unrecognized methane sink that reduces methane emissions in salt dome settings perhaps associated with an extensive, deep subsurface biosphere.

[1] Department of Geological Sciences, California State University, Fullerton, 800 North State College Boulevard, Fullerton, CA 92831, USA. [2] Department of Geological Sciences, University of Texas at Austin, 2275 Speedway Stop C9000, Austin, TX 78712, USA. [3] Department of Earth Sciences, University of California, Riverside, 900 University Avenue, Riverside, CA 92521, USA. [4] Department of Earth, Space and Planetary Sciences, Department of Atmospheric and Oceanic Sciences, Institute of the Environment and Sustainability, University of California, Los Angeles, 595 Charles Young Drive, Los Angeles, CA 90095, USA. Correspondence and requests for materials should be addressed to S.J.L. (email: sloyd@fullerton.edu)

Earth's climate is modulated by the concentration of atmospheric greenhouse gases. These gases (e.g., $CO_2$ and $CH_4$) are largely generated at depth and subsequently transported to surface environments. These gases may fail to reach the surface due to chemical reaction along the way, resulting in their degradation and the subsequent precipitation of mineral phases. The geochemical compositions of these diagenetic minerals can provide insight into the nature of degradation mechanisms, thus leading to a better understanding of the fate of subsurface gaseous and aqueous chemical species.

Evaporites, elemental sulfur, metal sulfides, and carbonate minerals can co-occur in unique diagenetic settings. Examples of this association include those observed in cap rocks formed atop Jurassic salt in the Gulf of Mexico Basin (GMB)[1], Permian salt of Germany and the North Sea Basin[2], and Triassic salt in northern Tunisia[3]. Similar deposits occur in Permian, hydrocarbon-bearing evaporite successions of the Delaware Basin in western Texas and Miocene strata in Carpathian basins of Poland, Ukraine, and Iraq[4]. These systems have been studied extensively due to their association with economic hydrocarbon and mineral resources[5]. Such environments provide appropriate conditions for microbial communities to take advantage of mineral-, aqueous-, and hydrocarbon-sourced reactants for metabolic gain. Despite the likelihood of active microbial cycling, little is known about the specific natures and impacts of these interactions.

The GMB subsurface represents one of the world's best-developed salt dome provinces (Fig. 1), containing hundreds of salt structures associated with post-depositional diapiric movement of the Jurassic Louann Salt[6-8]. The Louann Salt consists primarily of halite with minor (1–5%) anhydrite and gypsum[9]. Dome structures form when salt mobilizes and intrudes into overlying strata, partially as a result of the preferential subsidence of surrounding sediments. The GMB is also known for its large deposits of oil and natural gas that typically accumulate along the flanks of salt domes as a result of confinement by structural traps[10].

Approximately 65% of the onshore salt domes in the GMB are mantled by thick (up to 300 m) cap rock[8]. This cap rock consists of anhydrite, gypsum, and carbonate that can exhibit complex intergrown habits (Fig. 2)[8,11]. The cap rock is thought to form via hydrological, chemical, and microbial interactions[8,11,12] in the following generalized paragenetic sequence (also see Fig. 3).

Initially, salt migrates towards the surface and the characteristic domed structure forms (Fig. 3, step 1)[12]. Hydrocarbon traps develop along dome flanks as strata are deformed into on-lapping and anticlinal features. Off-flank reservoirs can also form in response to dynamic interrelations of sedimentation and salt withdrawal. The upper portions of the dome interact with meteoric or marine fluids, promoting halite dissolution. As halite dissolves, the less soluble anhydrite and gypsum accumulate as a residuum on the dome crest (Fig. 3, step 2).

Through continued interaction with undersaturated fluids, the accumulated anhydrite and gypsum dissolves, releasing calcium and sulfate into pore waters (Fig. 3, step 3). Hydrocarbon species react with dissolved sulfate, resulting in increased concentrations of dissolved inorganic carbon (DIC), alkalinity, and sulfide (although some sulfide may originate from deeper basinal brines[13]). Calcium reacts with produced DIC, promoting formation of authigenic carbonate minerals (Fig. 3, step 4). Aqueous sulfide can react with oxidants to form elemental sulfur or with divalent metals to produce sulfide minerals. These processes tend to produce a unique spatial relationship among cap rock phases, where older precipitates broadly occur stratigraphically above younger layers (termed inverted stratigraphy)[14]. Ultimately, this paragenetic description represents a generalized sequence of events.

Despite this longstanding model for cap rock formation, the carbonate mineral-forming reactions remain poorly characterized. Of particular importance is the identification of specific microbial reaction pathways, many of which are known to yield alkalinity and promote carbonate precipitation[15]. Early studies report carbonate $\delta^{13}C$ values ($\delta^{13}C_{carb}$) that range from −54‰ to −2‰ VPDB (Vienna-Pee Dee Belemnite)[11,16], indicating precipitation from a carbon source depleted in $^{13}C$ relative to seawater[8]. It was first proposed that carbonate carbon was sourced solely from liquid hydrocarbon due to its direct association with salt domes[11,17]. Indeed, liquid hydrocarbon can promote carbonate precipitation via alkalinity production through microbially mediated or high temperature (>100 °C)[18] reactions coupled with sulfate:

$$2CH_2O + SO_4^{2-} \rightarrow 2HCO_3^- + H_2S, \qquad (1)$$

where $CH_2O$ represents simplified liquid hydrocarbon. However, the $\delta^{13}C$ values of modern GMB oils are relatively narrow, ranging from −28.0 to −23.0‰[19]. In the GMB subsurface, methane represents the only carbon source that is sufficiently $^{13}C$-depleted to produce $\delta^{13}C_{carb}$ below −28‰[8,20]. Methane can act as a reducing agent during microbial sulfate reduction

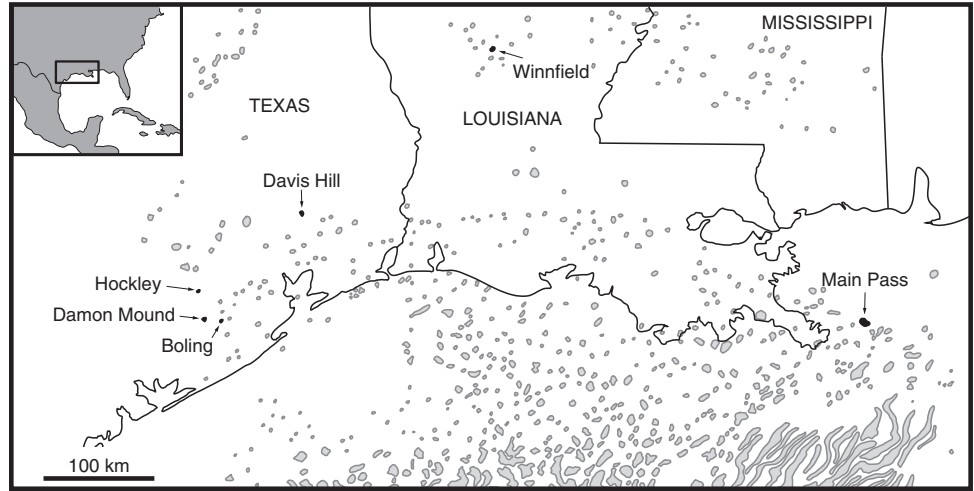

**Fig. 1** Distribution of northern Gulf of Mexico Basin salt structures. Salt domes studied here are highlighted in black. Map modified from Martin[71]. AAPG©1978 reprinted by permission of the AAPG whose permission is required for further use

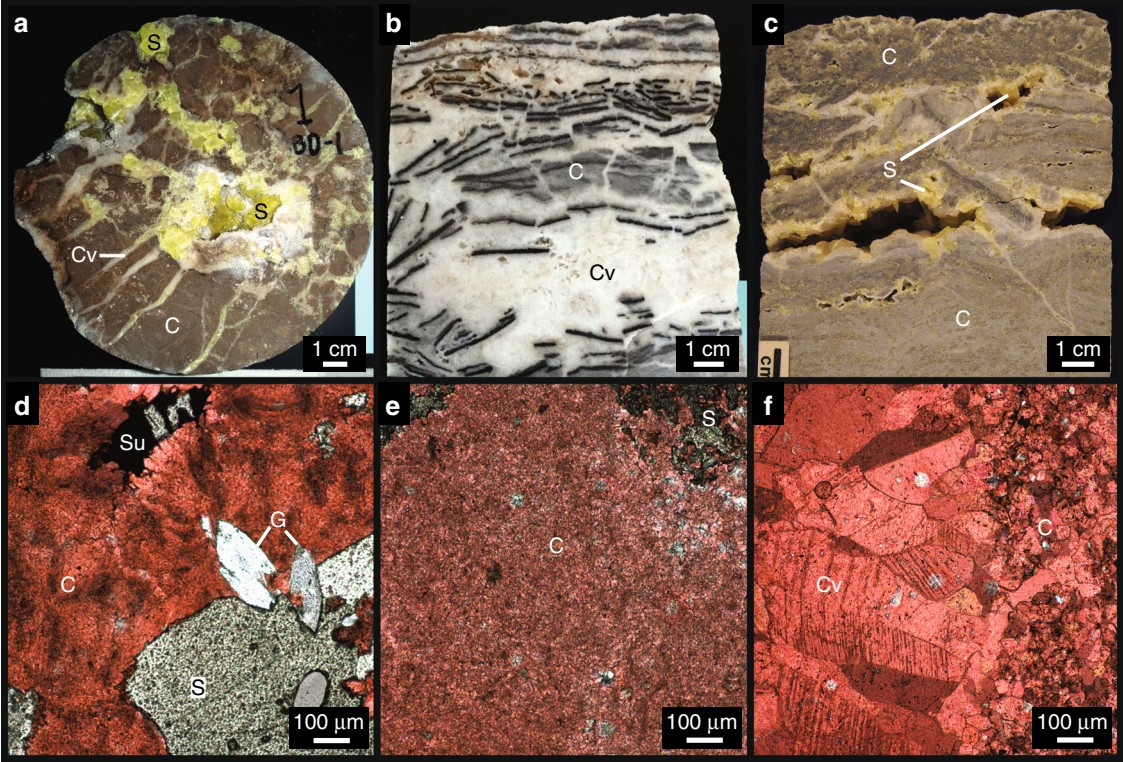

**Fig. 2** Photographs and photomicrographs of Gulf of Mexico Basin cap rock. Samples from Boling Dome (**a**, **d**), Winnfield Dome (**b**, **e**), and Main Pass Dome (**c**, **f**) are pictured. Gypsum (G), host carbonate (C), vein carbonate (Cv), yellow elemental sulfur (S) and black metal sulfide (Su) minerals are clearly visible in hand sample and thin section. Red stain in **d**–**f** indicates calcium carbonate

through sulfate-dependent anaerobic oxidation of methane (AOM)[21,22], in accordance with the following net reaction:

$$CH_4 + SO_4^{2-} \rightarrow HCO_3^- + HS^- + H_2O. \qquad (2)$$

Microbial anaerobic oxidation of methane has been shown to promote extensive authigenic carbonate formation on the seafloor and in shallow marine sediments[23–25]. A similar reaction occurs at relatively high temperatures in the absence of microorganisms[18].

Solid sulfur-bearing phases occur predominantly in GMB cap rock as sulfates, sulfides, and elemental sulfur[8]. Previous studies indicate that cap rock sulfur isotope values vary significantly among these phases, ranging from −40 to +78‰ VCDT (Vienna-Canyon Diablo Troilite)[13]. In general, sulfides and elemental sulfur in northern GMB cap rock express [34]S-depleted isotope compositions relative to the gypsum and anhydrite inclusions contained within the parent Louann Salt. In contrast, cap rock sulfate-bearing barite and celestine (SrSO₄) commonly exhibit [34]S-enriched values[8]. This sulfur isotope variability typifies microbially dominated systems, as large isotope fractionations can result from microbial sulfate reduction and/or disproportionation[26]. Despite evidence for microbial cycling in salt dome settings, relationships between carbonate mineral production and these biogeochemical reactions remain poorly understood.

Here, we explore GMB cap rock from six domes across Texas and Louisiana: Boling, Davis Hill, Damon Mound, Hockley, Main Pass, and Winnfield domes (Fig. 1). Petrographic and isotopic data from carbonate and sulfur phases have been collected to better constrain carbonate mineralization pathways. These new data along with data provided in the literature support a relatively low-temperature, microbial precipitation mechanism

that includes sulfate- and hydrocarbon-based reactants, as discussed in detail below.

## Results and Discussion

**Cap rock paragenesis**. Previously reported carbon and sulfur isotope data[8,11,13,16,17,27] indicate potential for hydrocarbon oxidation-linked sulfate reduction as a cap rock mineral precipitation mechanism. However, these data derive from different proxy reservoirs (carbonate mineral δ[13]C and sulfur mineral δ[34]S), which may have formed during distinct (and perhaps unrelated) diagenetic events. In fact, it has been proposed that barite and celestine (sulfate minerals that exhibit [34]S enrichments consistent with sulfate reduction under sulfate-limited conditions) formed during relatively late stages of cap rock development[8].

Of the six domes explored here, four exhibit cap rock with significant elemental sulfur and sulfide mineral accumulations: Boling, Davis Hill, Hockley, and Main Pass domes. These sulfur phases exhibit variable paragenetic relationships with carbonate (Fig. 4). Carbonate occurs mostly as micritic, microspar, and spar cements. Later-stage vein precipitates commonly cross-cut early micrtitc and microspar cements[8] (Fig. 2b). In many instances, carbonate phases formed after anhydrite and gypsum, as indicated by pseudomorphic replacement. Petrographic examination reveals that carbonate cements formed before, contemporaneous with, and/or after elemental sulfur and sulfide phases depending on the locality. Elemental sulfur in Boling Dome cap rock mineralized before and after carbonate, as indicated by cross-cutting relationships (Fig. 4a, b). Radiating sulfide crystals mimic carbonate (likely aragonite given the fibrous habit), perhaps reflecting contemporaneous or subsequent sulfide precipitation (Fig. 4c). Sulfide minerals at Davis Hill Dome tend to fill veins and cross-cut carbonate microspar cement, indicating sulfide mineralization after carbonate (Fig. 4d).

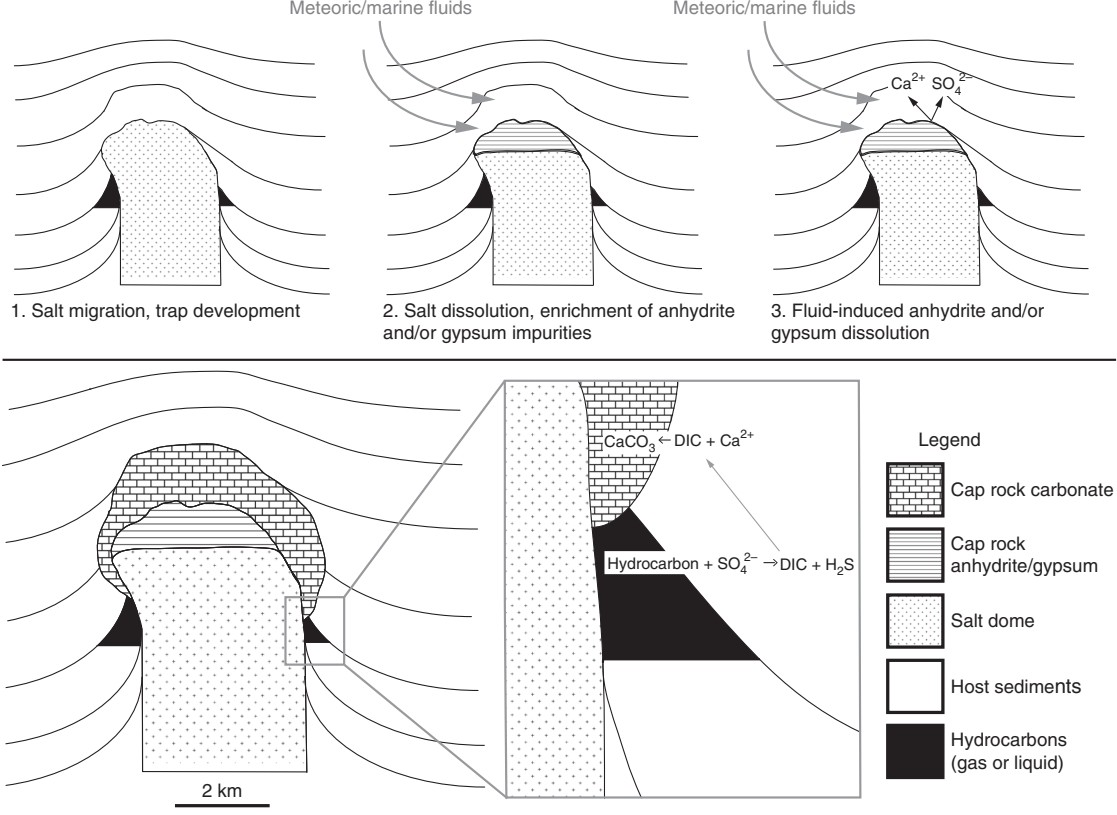

**Fig. 3** Proposed paragenetic evolution of Gulf of Mexico Basin cap rock. The specific nature of degraded hydrocarbon is largely uncharacterized, as is the relationship between degradation pathway and carbonate authigenesis (see step 4)

Sulfide minerals within cap rock of Hockley Dome display disseminated (Fig. 4e) and botryoidal crystal habits (Fig. 4f, g). The disseminated habit may reflect precipitation contemporaneous with carbonate formation. Where botryoids dominate, carbonate cement crosscuts (Fig. 4f) and/or nucleates on sulfide (Fig. 4g), indicating that carbonate precipitated after sulfide. Elemental sulfur within cap rock of Main Pass Dome often crosscuts carbonate cement (Fig. 4h). Sulfide occurs as void-filling precipitates, with carbonate crystal terminations extending into the sulfide (Fig. 4i). These relationships indicate that both elemental sulfur and sulfide formed after carbonate.

Although in some cases petrographic relationships indicate variable formation timing between cap rock carbonate and sulfur phases, it has been proposed that these phases can form within a relatively narrow time window[27]. When considered as a whole, however, complex paragenetic relationships (and an overall lack of phases that conclusively formed contemporaneously with carbonate) can confound interpretations regarding carbonate mineralization pathways. Ultimately, carbon and sulfur isotope data hosted in the same carbonate minerals will provide insight into the sulfur–carbon reactions that led to carbonate mineralization.

**Cap rock carbonate mineralization mechanism.** Our new $\delta^{13}C_{carb}$ data vary widely, ranging from −52.7 to −2.9‰ (Fig. 5), consistent with previous reports[11,16]. Most of these values fall below modern GMB petroleum (with a minimum $\delta^{13}C$ of −28‰), likely indicating a significant carbon contribution from the oxidation of methane. In some cases, microbial oxidation of organic matter is accompanied by an isotopic depletion in the produced dissolved inorganic carbon. In general, however, this isotopic fractionation is small, leading to $^{13}C$-depletions in

DIC of ~3‰ or less[28–30] (although acetate oxidation reactions can yield more severe fractionations[31]). Therefore, carbonate $\delta^{13}C$ values that are less than ~−28‰ likely indicate a methane carbon source. Cap rock that is less depleted in $^{13}C$ (heavier than −28‰) may have received carbon from the oxidation of liquid hydrocarbon or from the dissolution of marine limestone, both of which are abundant in the GMB subsurface. However, $\delta^{13}C_{carb}$ values greater than the liquid hydrocarbon minimum do not preclude a contribution from methane-derived carbon but rather suggest that carbon was provided from multiple sources (as has been recognized in marine cold seep settings[32]).

Information about contemporaneous sulfur cycling can be obtained through the isotope composition of carbonate-associated sulfate ($\delta^{34}S_{CAS}$). Carbonate-associated sulfate (CAS), trace sulfate incorporated into the carbonate mineral lattice upon precipitation, has been shown to record the sulfur isotope composition of ambient aqueous sulfate[33,34], including that of diagenetic systems[35,36]. Given that CAS is incorporated as sulfate, it provides a unique opportunity to explore the isotopic evolution of the most oxidized end-member of sulfur. As a carbonate-hosted proxy, $\delta^{34}S_{CAS}$ can be used in tandem with $\delta^{13}C_{carb}$ to elucidate coupled carbon–sulfur reaction pathways and relationships to carbonate mineral precipitation.

$\delta^{34}S_{CAS}$ values from host and vein cap rock carbonates range from +12.5 to +68.8‰ (Fig. 5). Dominantly $^{34}S$-enriched values (compared to the source sulfate derived from the Louann Salt, with a $\delta^{34}S$ value of ~+16‰) indicate microbial sulfate reduction under sulfate-limited conditions[37]. As bacteria preferentially reduce $^{32}SO_4^{2-}$, the residual sulfate pool experiences progressive enrichment in $^{34}S^{37}$ and consequently yields relatively high $\delta^{34}S_{CAS}$ values in contemporaneously precipitated carbonate. Preferential reduction of $^{32}SO_4^{2-}$ to sulfide can promote the

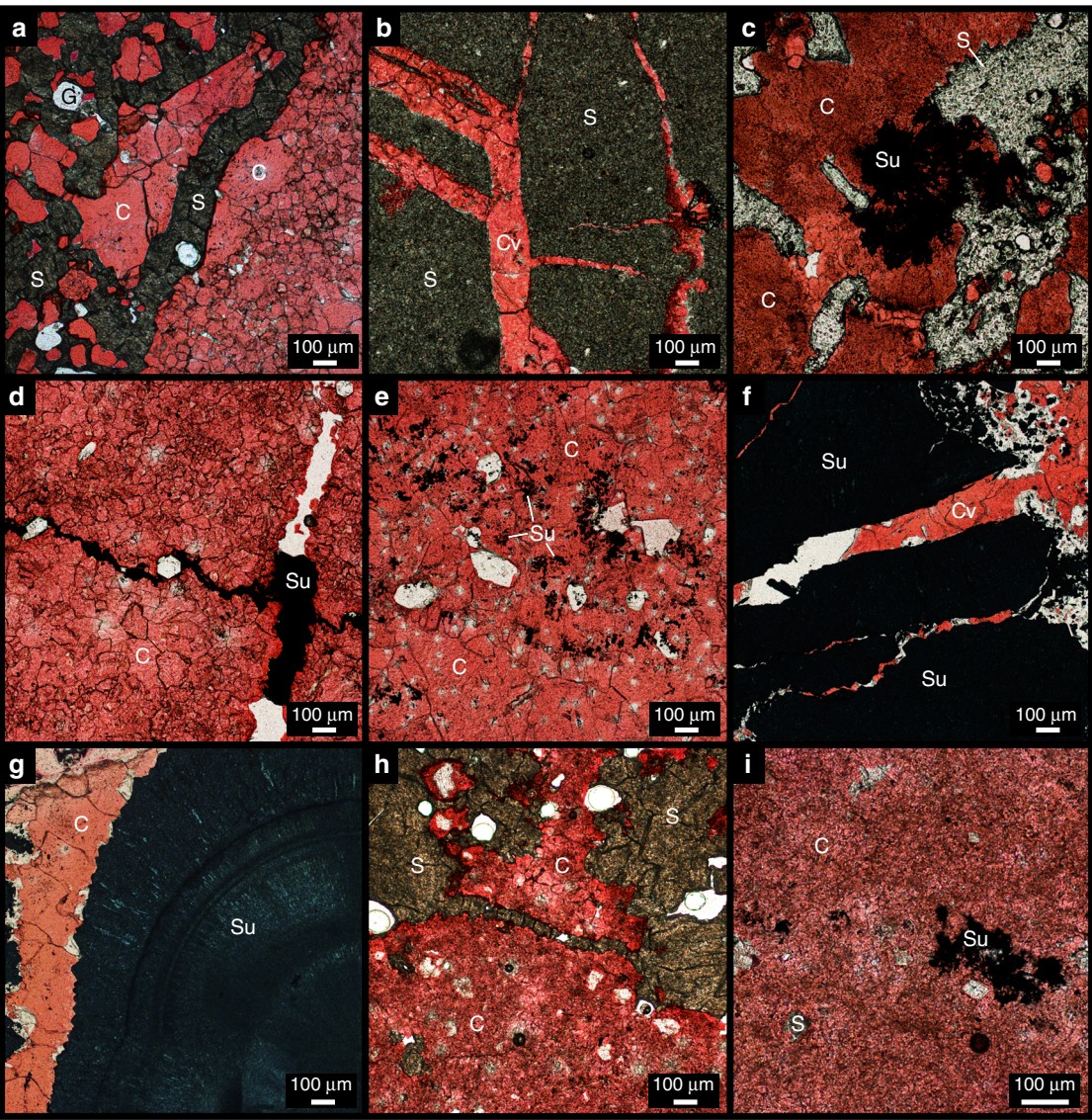

**Fig. 4** Photomicrographs displaying paragenetic relationships between reduced sulfur phases (elemental sulfur and sulfide) and carbonate. **a–c** Boling Dome, **d** Davis Hill Dome, **e–g** Hockley Dome, and **h**, **i** Main Pass Dome. Labels correspond to Gypsum (G), host carbonate (C), vein carbonate (Cv), yellow elemental sulfur (S), and metal sulfide (Su) phases. Red stain indicates calcium carbonate

production of $^{34}$S-depleted reduced sulfur mineral phases (Fig. 5b). With progressive sulfate depletion, however, these sulfides can also exhibit relative $^{34}$S enrichment as the parent sulfate pool evolves. The common occurrence of metal sulfide $\delta^{34}$S values near 0‰ may indicate mineralization after the development of sulfate-limited conditions[27]. Ultimately, $^{13}$C-depleted carbonate carbon and $^{34}$S-enriched CAS suggests simultaneous oxidation of methane and reduction of sulfate and thus that sulfate-dependent AOM (Eq. 2) promoted cap rock carbonate precipitation.

In many instances it can be difficult to distinguish between microbial and thermochemical reaction pathways using traditional carbon and sulfur isotope data alone. Thermochemical sulfate reduction, however, is limited to high-temperature environments (above ~100 °C)[18]. Clumped isotope compositions (reported as $\Delta_{47}$, see below) have proven useful in distinguishing carbonate mineralization temperatures in both primary and diagenetic settings. The $\Delta_{47}$ values of GMB cap rock carbonates range from 0.585 and 0.720‰ (reported in the absolute reference frame (ARF)[38]), suggesting carbonate precipitation temperatures between ~26 and 83 °C[39,40], depending on the calibration.

These temperatures fall well below the lower limit for thermochemical sulfate reduction (Fig. 6), in agreement with the common occurrence of single-phase fluid inclusions within calcite at these and other localities[41]. Modern marine cold seep carbonates precipitated via AOM yield anomalously low $\Delta_{47}$ compositions (likely inherited through kinetic effects), corresponding to precipitation temperatures significantly above ambient conditions[42]. Since these kinetic effects consistently yield temperature estimates higher than ambient conditions, we can expect the same for GMB carbonates. Specifically, it is likely that our temperatures are also overestimates and thus are well below those required for thermochemical sulfate reduction. Barite fluid inclusion data from Hazlehurst Salt Dome also indicate relatively low formation temperatures (<55 °C)[27]. In addition, samples recording the maximum degree of $^{34}$S-depletion in elemental sulfur and sulfide minerals preserved in cap rocks of the GMB are difficult to explain through thermochemical reactions alone, which generally exhibit maximum isotope discriminations of ~20‰[18]. Thus, temperature proxy data derived from both carbonate and sulfur phases indicate temperatures that are too low to facilitate thermochemical sulfate

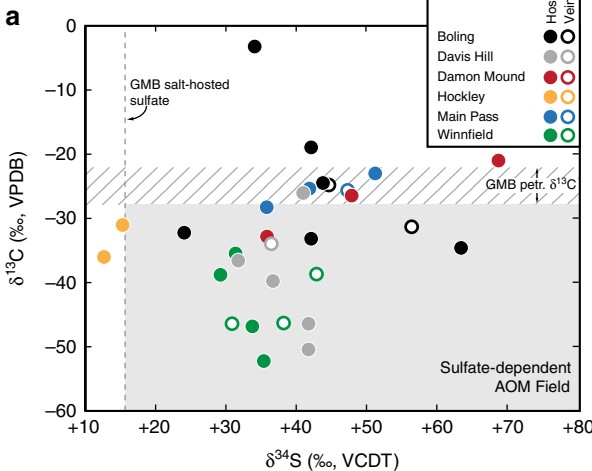

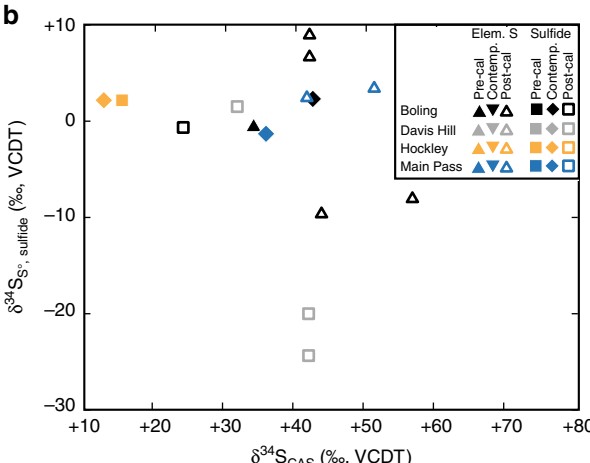

**Fig. 5** Carbon and sulfur isotope data of Gulf of Mexico Basin (GMB) cap rock phases. **a** Carbonate carbon ($\delta^{13}C_{carb}$) and sulfur isotope ($\delta^{34}S_{CAS}$) compositions of cap rock carbonate. Early (host) and late (vein) phases are identified. Also shown are the $\delta^{13}C$ ranges for northern GMB petroleum (hatched envelope) and the $\delta^{34}S$ value for minor anhydrite/gypsum in the Louann Salt (vertical dashed line). Carbonate carbon isotope composition lower than –28‰ and carbonate-associated sulfate (CAS) sulfur isotope compositions above +16‰ require sulfate-dependent anaerobic oxidation of methane as a mineralization mechanism. **b** Comparative reduced sulfur and carbonate-associated sulfate isotope data. Paragenetic relationship with carbonate indicated by symbol. In both panels, symbols are larger than the error for each measurement

reduction. These findings support the hypothesis that the AOM reactions identified through integrated geochemical analyses of salt dome cap rock were most likely mediated by microorganisms.

Microbial AOM reactions occur in modern seafloor environments experiencing methane seepage and lead to the formation of extensive authigenic carbonate and sulfide phases[24,25]. During AOM, a consortium of methanotrophic and sulfate-reducing microbes consume methane and sulfate and produce DIC and dissolved sulfide[21,22]. This process leads to conversion of methane carbon into bicarbonate via oxidation, thereby increasing pore water DIC and alkalinity (promoting the formation of authigenic carbonate) and decreasing fluid $\delta^{13}C$ values[24]. Reactions involving the oxidation of petroleum may also be important sources of alkalinity, although the higher $\delta^{13}C_{carb}$ data reported here do not conclusively indicate petroleum oxidation, as some carbon may also be sourced from the dissolution of $^{13}C$-enriched marine

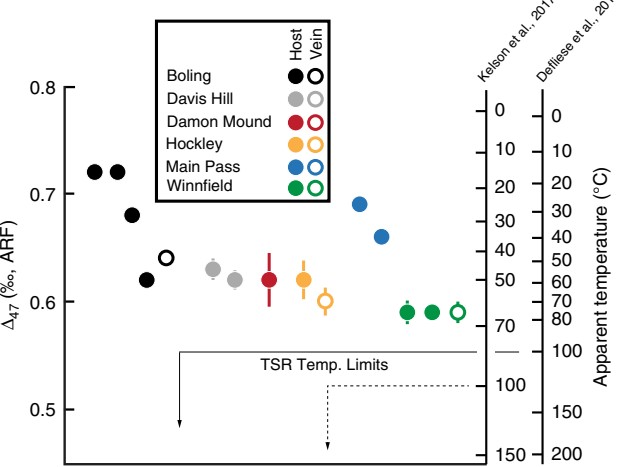

**Fig. 6** Clumped isotope data from Gulf of Mexico Basin cap rock carbonate. Notice that in all cases, calculated temperatures fall below the thermochemical sulfate reduction (TSR) lower limit. Error bars correspond to ±2 standard errors (s.e.)

limestone. In contrast, $^{13}C$-depletions below –28‰ are indicative of methane oxidation (Fig. 5a). In salt dome environments, both methane and petroleum oxidation may foster the growth of carbonate cap rock, although in this case (as opposed to marine cold seeps, for example) both calcium and sulfate are derived from the dissolution of sulfate minerals (anhydrite and gypsum) rather than seawater. Ultimately, without the increase in alkalinity generated through sulfate-dependent AOM and petroleum oxidation, the production of carbonate in cap rock would be less likely.

Some salt domes lack carbonate cap rock. Several Permian Zechstein salt diapirs in Germany and Poland exhibit only gypsum cap rock and lack associated hydrocarbons, carbonate, and elemental sulfur[2,43]. In these settings, $\delta^{34}S$ values of dissolved sulfate in local aquifers indicate closed-system sulfate reduction, perhaps coupled with organic matter oxidation[44] rather than with the oxidation of petroleum or methane. Despite the occurrence of sulfate reduction, authigenic carbonates are absent[44,45]. This relationship may indicate that methane- and/or petroleum-coupled sulfate reduction is/are necessary for extensive formation of cap rock carbonate in salt-related diagenetic environments.

The above discussion (and Eq. 2) treats AOM as a sulfate-driven chemical reaction. However, anaerobic methane oxidation has been shown to occur through reaction with nitrate[46], iron oxides, and manganese oxides[47] and is thermodynamically possible through other reaction pathways as well[48]. Whereas most AOM is restricted to marine environments due to the abundance of sulfate in seawater, terrestrial and fresh water systems can exhibit both sulfate- and non-sulfate-dependent AOM[49–51]. In addition, it has been shown that some deep subsurface microbial ecosystems can facilitate AOM[52–54] and promote the precipitation of diagenetic carbonate[51], a process that may have occurred throughout the Phanerozoic[55]. Here, we demonstrate that salt dome basins represent unique, non-marine, subsurface environments wherein sulfate-dependent AOM occurs and leads to the production of diagenetic carbonate and other minerals. This AOM is promoted through complex hydrological interactions that yield aqueous sulfate through calcium sulfate mineral dissolution[12]. Similar sulfate-dependent AOM may occur in other environments where sulfate-rich minerals experience dissolution in proximity to significant methane accumulations, such as those produced in some evaporite settings[56].

**Methane consumption in the salt dome environment**. Methane has played an important role in global climate throughout Earth history and remains a significant contributor to greenhouse forcing today. It has been estimated that AOM reduces the emission of methane derived from marine sediments by ~80%[57,58]. Without this oxidation mechanism, it is projected that atmospheric $CH_4$ levels would be 10–60% higher[59–61]. Mediating methane escape is especially important in the GMB region, where natural methane and petroleum seeps are abundant[62,63]. Salt migration creates extensive fault networks that serve as conduits whereby methane and petroleum may escape into the atmosphere and ocean[62]. AOM occurring in the GMB in association with salt domes may reduce methane emissions from these natural seeps[64], perhaps serving as an important and unrecognized methane sink.

Ultimately, it is difficult to accurately determine the degree to which subsurface oxidation limits methane escape from the GMB subsurface. Now, we will attempt to broadly constrain the quantity and rate of methane oxidation by extension of volume data for cap rock from one of the best-studied GMB domes, Boling Dome (spatial and volumetric data from Samuelson[65] and Kyle[66]). The maximum thickness of carbonate cap rock from Boling Dome is ~120 m. If we consider an average thickness of 60 m over the approximate area of the dome (~$3.9 \times 10^7$ m$^2$ at the 1220 m depth) and an approximate porosity of 15%, the total volume of carbonate cap rock is ~$1.99 \times 10^9$ m$^3$, and ~$5.4 \times 10^{13}$ moles of carbon are preserved as cap rock carbonate in association with Boling Dome. It has been estimated that 65% of the 500 onshore GMB salt domes exhibit cap rock[8], suggesting that a total of ~$1.8 \times 10^{16}$ moles of carbon are preserved as cap rock carbonate throughout the basin. If half of this carbon derived from methane oxidation (as is approximated by the range of $\delta^{13}C$ data, Fig. 5a), then subsurface oxidation accounts for the consumption of ~$9.0 \times 10^{15}$ moles of $CH_4$.

It is even more difficult to constrain the rate of methane consumption associated with cap rock reactions. This challenge is largely the result of our inability to accurately determine cap rock ages and formation rates. Theoretically, cap rock formation may have initiated any time after Louann Salt deposition (~165 Ma).

The global net rate of methane growth in the atmosphere has been approximated at ~6 Tg $CH_4$/year[67]. In order for salt dome-associated methane oxidation to decrease this growth rate by ~1% requires formation of all onshore GMB cap rock carbonate over ~2.3 Myr (see Supplementary Figure 1, also includes sensitivity to methane carbon contributions). This estimate does not take into account the likelihood that some of the AOM-produced DIC may not have precipitated as carbonate or the possibility of carbonate dissolution. Indeed, the potentially very old (up to ~165 million years) age of cap rock provides ample opportunity to change the subsurface chemical environment to conditions that favor carbonate dissolution. Furthermore, many additional salt domes are present in offshore settings relative to onshore or nearshore continental shelves of the GMB (Fig. 1). The presence of carbonate cap rock in these offshore locales has been documented[68], but the extent to which cap rock is developed is not well constrained. Ultimately, the AOM rate may not be (or have been) globally significant; however, the amount of carbonate produced as a result of this process has led to widespread mineral accumulation. These accumulations are so extensive that they have been commercially mined from economic use[5].

The data of this study reveal that cap rocks in the GMB formed at relatively low temperatures as a product of substantial microbial methane oxidation. The results further suggest that such processes may mitigate against methane release to the atmosphere, although the global impact (as compared to modern methane growth rates) might be minimal. Cap rock carbonate represents a potentially unique biosignature recording extensive microbial activity within the deep subsurface.

## Methods

**Carbon isotope analyses**. Samples were acquired from six salt domes across Texas, Louisiana, and offshore Gulf of Mexico (Fig. 1). Samples were prepared and analyzed for $\delta^{13}C$ at California State University, Fullerton. Then, 5 mg of powdered sample was weighed out and placed into borosilicate glass Exetainer® vials. Atmosphere was evacuated by vacuum and 3 mL of 10% phosphoric acid was added to each vial to acidify carbonate and release $CO_2$ (g). The samples were allowed to react overnight. Produced $CO_2$ was passed via ultra-high purity nitrogen carrier gas into a G2121-i Picarro Inc. Cavity Ringdown Spectrometer (CRDS) isotope analyzer through an Automate © Carbonate Prep Device. Carbonate isotope data are reported in permil (‰) using the standard δ-notation relative to the VPDB standard. Reproducibility is better than 0.5‰ (2 s.d.).

**Sulfur**. Extraction of cap rock CAS was performed following standard methods[33,36]. Samples were first cut into billets and then ground to a powder. Approximately 15 to 50 g of powdered samples were rinsed in 1 L of 10% NaCl solution in order to remove soluble sulfur species (gypsum and anhydrite). The mixture was allowed to sit for ~8 h, and the supernatant fluid was siphoned from the flask and discarded. The NaCl wash was then repeated. After two NaCl washes, 1 L of 5% bleach solution was added and allowed to sit for ~8 h in order to remove any organic-bound sulfur species. After 8 h, the supernatant fluid was siphoned from the flask and discarded. Two additional NaCl washes followed the bleach step. To liberate the carbonate-associated sulfate as aqueous sulfate, 500 mL of 6 M HCl was slowly added to each sample and swirled. The flasks were then left overnight to ensure complete sample dissolution. The fluid was then filtered successively through 40, 11, and 0.45 μm pore diameter filters. Filtered fluids were then heated to ~80 °C, and a 30% $BaCl_2$ solution was added. These samples were allowed to sit for 72 h to allow aqueous sulfate to be precipitated as barite. The solution was then filtered through a preweighed 0.45 μm filter using a vacuum-assisted flask to isolate the fluid from the barite. These filters were then dried for approximately 4 h at 80 °C.

Elemental sulfur and sulfide sulfur were collected by microdrilling with a Dremel rotary tool fitted with a 1-mm diameter carbide drill bit. Approximately 6–8 mg of elemental sulfur and 10–20 mg of metal sulfide were subjected to chromium reduction according to traditional methods[69]. Samples were heated and reacted with a 1 M $CrCl_2$/HCl and ethanol solution under a $N_2$ atmosphere. This reaction converted elemental sulfur and metal sulfide to gaseous $H_2S$ and the product gas was passed into a trap containing 50 mL of a 3% $AgNO_3$/10% $NH_4OH$ solution. Reaction of sulfide gas with the trap solution yielded solid silver sulfide, which was used as analyte for isotope composition determination.

Sulfur isotope analyses of CAS-extraction-produced barite and chromium-reduction-produced silver sulfide were conducted on a ThermoScientific Delta V Plus IRMS at the University of California, Riverside. The IRMS is connected to a Costech Analytical Technologies, Inc., elemental combustion system via a ThermoScientific CONFLO III interface. Samples were combusted in tin capsules with 2.000 mg (±0.500) of vanadium pentoxide ($V_2O_5$), added as a catalyst to ensure complete combustion. Sulfur isotope values are reported in ‰ using δ-notation relative to the VCDT standard. Replicate $\delta^{34}S$ data are generally better than ±1.0‰ (2 s.d.).

**Clumped isotope analyses**. Select samples were microdrilled with a 1 mm carbide drill bit and approximately 50 mg collected for clumped isotope analyses. Then, 6–10 mg of powder was acidified with supersaturated phosphoric acid and the generated $CO_2$ was passed through a series of cryogenic traps using an on-line automated preparation device (similar to that described in Passey et al.[70]) with digestion and clean-up methods identical to Loyd et al.[42]. The abundance of mass 47 $CO_2$ was determined using a Thermo MAT 253 gas source isotope ratio mass spectrometer and reported in the conventional $\Delta_{47}$ notation in ‰ after being cast into the ARF[38]. Isotopic ratios are calculated using the Brand parameter set and standardization utilized both equilibrated gases and carbonate standards. These $\Delta_{47}$ values were converted to temperature using low[40] and high[39] temperature end-members. Replicate $\Delta_{47}$ data were generally better than ±0.013‰ (1 s.e.).

## Data availability

All data pertinent to this study are provided in the manuscript.

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

## Acknowledgements

This research was partially supported by the Donors of the American Chemical Society Petroleum Research Fund (Grant #55428-UNI2, awarded to S.J.L.). Additional support was provided by the III C.E. Yager professorship (to J.R.K), by CSUF Incentive (Loyd 2014) and Jr. Intramural (#03540) grants awarded to S.J.L. and a Geological Society of America Graduate Student grant awarded to K.H.C. A.T. was supported by the Department of Energy through BES grant DE-FG02-13ER16402.

## Author contributions

S.J.L. conceived project, J.R.K. provided samples and geologic context, K.H.C. conducted lab work, and A.T. conducted clumped isotope analyses. K.H.C., J.R.K., T.W.L., A.T. and S.J.L. contributed to data analysis, interpretation, and manuscript drafting.

## Additional information

**Competing interests:** The authors declare no competing interests.

