## [Peer Review File · Nature Communications]

Reviewers' comments:

Reviewer #1 (Remarks to the Author):

This is a potentially interesting manuscript about anaerobic oxidation of methane (AOM) associated with salt domes (cap rocks). The authors claim that carbonates formed as cap rocks above these salt domes serve as unrecognized widespread sinks for hydrocarbons. The authors use mainly C and S isotopes in carbonate and carbon associated sulfate, respectively, to prove their point. Whereas the C isotope data indeed speaks in favor of an AOM process, the S isotope data and interpretations are more problematic. In its current form the manuscript does not convincingly prove microbial sulfate reduction or rule out thermochemical sulfate reduction. S isotope data are given and cited, but are not presented conclusively or in such detail that a full process understanding can be deduced. I have suggested a couple of avenues (below) that the authors could follow to prove their point more convincingly. Furthermore, much more information about the system is needed (i.e. geological setting, textural documentation) to make the story comprehensive. As an example, the study is on carbonate minerals, but it is unclear whether it is a fully fossil system or feature ongoing processes. The system seems to have evolved over time. Are there any age constraints on these different phases and/or on the process as a whole? It is also not clear why only methane oxidation is able to form carbonate cap rocks when other processes can produce bicarbonate as well and other carbon sources are indicated in the isotope data of the carbonates (not all carbonate-C is ^{13}C -depleted). Substantial revision and addition of analytical data (see bullets below), sample descriptions (preferentially the relation between pyrite and calcite) and geological setting are needed before it can be recommended for publication. It is recommended to discuss and compare the findings in the context of other recently described non-marine methane sources and sinks (such as those described in various works by Etiope and colleagues (Etiope & Sherwood Lollar, 2013 ; Etiope, 2009 ; Etiope & Klusman, 2002) and the similarly operating carbonate mineral sink for methane recently discovered in the upper continental crust (Drake et al., 2017 ; Drake et al., 2015 ; Sahlstedt et al., 2016), which also have been discussed from microbiological perspectives (Ino et al., 2017 ; Bomberg et al., 2015 ; Simkus et al., 2016) and not only comparing the data with marine findings. To conclude, the manuscript is interesting and the recognition of a widespread methane sink in salt dome cap rocks has wide implications, but the methods used should be expanded and the data more carefully presented/interpreted in order to be in line with the high standard of Nature Communications. The journal has quite generous word limit allowing the authors to add more details.

Specific remarks:

1. As presented, the S-isotope data seems to be used as a diagnostic tool for AOM. For instance, there is a field in fig.4 "AOM Field", but strictly speaking there are no specific S-isotope values that are diagnostic for AOM. Yes, S-isotopes can be used to prove microbial sulfate reduction (MSR) but not independently to prove AOM. This is also indicated in the text at lines 246-249 describing similar range of S-isotope values from diapirs without methane oxidation.
2. The heavy S isotope values are used to prove MSR in closed system. However, there are a wide range of values that needs to be discussed to understand the whole system. Heavy $\delta^{34}\text{S}$ values can be due to closed system Rayleigh fractionation but then what about MSR that is not undergoing closed system fractionation? This cannot be distinguished using sulfate $\delta^{34}\text{S}$ only but might still be occurring. A much more straightforward way to gain knowledge of MSR is to study the product of MSR (i.e. sulfide minerals) instead of the residual sulfate alone. By using micro-scale analysis in pyrite (e.g. SIMS) it is possible to distinguish the evolution of a closed-system within the setting (see e.g. Drake et al., 2015; Lin et al., 2016). If there was abundant MSR (related to AOM) there should also be pyrite present. The authors use $\delta^{34}\text{S}$ in pyrite from cited data but i) do not present it comprehensively (instead presenting it grouped with other S-minerals) and ii) state that it is not certain whether pyrite and calcite are co-genetic. So why used it at all in that case? It is recommended instead to do a thorough check in the samples if pyrite/calcite paragenesis can be distinguished and then analyze the pyrite for $\delta^{34}\text{S}$ (micro-scale analysis preferred).

3. d34S in CAS is heavy (at least partly, based on the presented data). This means that in order to have mass balance there must be significant amounts of isotopically light sulfide (or some other S-phase) precipitated in the system. The discussion should be expanded to include this mass balance problem.
4. The span in d34S values can theoretically be due to thermochemical sulfate reduction (TSR). This is ruled out by the authors due to that pyrite has very low d34S values (i.e. too large fractionation for abiotic processes). But there were, again, difficulties in assessing whether pyrite (cited data) and calcite were co-genetic so clearly the temporal pyrite-calcite relation needs to be strengthened and/or additional proof for MSR should be added. What about fluid inclusions in the calcites? That could prove that the calcites are low-T and rule out TSR.
5. It is not clear if methane was oxidized by microbes or result of abiotic processes. Fluid inclusion homogenization temperatures could support this interpretation as well, if the temperatures are low microbial methanotrophy is likely. In addition, it is well known from fossil AOM-MSR systems that organic remains are present within authigenic carbonates. There are several proxies that can be used (biomarkers, like specific fatty acids as well as isotope signatures of different compounds, see e.g. Niemann & Elvert, 2008; Elvert et al., 2003; Ziegenbalg et al., 2012)). The manuscript would be significantly strengthened by inclusion fluid inclusion analysis and/or analyses of any remains of microbial communities.
6. Statement that AOM is needed for carbonate cap rock formation. This is not fully supported by the data, because 1) the d13C values of the carbonate (cited and new data) can be as heavy as -2 permil which certainly is not related to AOM, so other sources of alkalinity exist. 2) there is evidently MSR from organic matter oxidation in salt dome/cap rocks as cited in the manuscript. This process could lead to HCO₃⁻ formation and hence in an increased alkalinity that would initiate carbonate precipitation. 3) Not all sites plot in the "AOM field" of Fig. 4. 4) The assumption that all calcite with values below -28 permil is shaky. There is fractionation to the lighter side when organic matter and oil are degraded by microbes (microbes use the lighter isotope preferentially), so the bicarbonate produced in this way can be lighter than -28 permil. There are clearly some borderline cases in that d13C range in fig 4. More discussion and more careful presentation on this is needed.
7. Addition of geochronological constraints would be useful.

Minor suggestions:

1. The process behind the abundant precipitation of elemental S should be described.
2. Describe more comprehensively in the introduction how the AOM+SR process works in general (based on findings e.g. at SMTZ in marine sediments and in continental crust).
3. Fig. 2, mark the mineral phases in the figure.
4. Line 151. Fractionation of d13C during AOM should also be taken into account, as well as mixing of C from AOM with other carbon sources, which taken together as a rule make d13C composition of authigenic carbonate different from the source methane (usually heavier methane, see Peckmann & Thiel, 2004).
5. Lines 164-172, describe the S-isotope data in greater detail by writing out values for different mineral phases and giving numerical information about what is meant by "large isotope fractionation".

References:

- Bomberg M, Nyyssönen M, Pitkänen P, Lehtinen A, Itävaara M (2015) Active Microbial Communities Inhabit Sulphate-Methane Interphase in Deep Bedrock Fracture Fluids in Oikiluoto, Finland. *BioMed Research International*, 979530.
- Drake H, Heim C, Roberts Nmw, Zack T, Tillberg M, Broman C, Ivarsson M, Whitehouse Mj, Åström Me (2017) Isotopic evidence for microbial production and consumption of methane in the upper continental crust throughout the Phanerozoic eon. *Earth and Planetary Science Letters*, 470, 108-118.
- Drake H, Åström Me, Heim C, Broman C, Åström J, Whitehouse Mj, Ivarsson M, Siljeström S, Sjövall P (2015) Extreme 13C-depletion of carbonates formed during oxidation of biogenic

methane in fractured granite. *Nature Communications*, 6, 7020.

Elvert M, Boetius A, Knittel K, Jorgensen Bb (2003) Characterization of specific membrane fatty acids as chemotaxonomic markers for sulfate-reducing bacteria involved in anaerobic oxidation of methane. *Geomicrobiology Journal*, 20, 403-419.

Etiopio G (2009) Natural emissions of methane from geological seepage in Europe. *Atmospheric Environment*, 43, 1430-1443.

Etiopio G, Klusman R (2002) Geologic emissions of methane to the atmosphere. *Chemosphere*, 49, 777-789.

Etiopio G, Sherwood Lollar B (2013) Abiotic methane on Earth. *Reviews of Geophysics*, 51, 276-299.

Ino K, Hermsdorf Aw, Konno U, Kouduka M, Yanagawa K, Kato S, Sunamura M, Hirota A, Togo Ys, Ito K, Fukuda A, Iwatsuki T, Mizuno T, Komatsu Dd, Tsunogai U, Ishimura T, Amano Y, Thomas Bc, Banfield Jf, Suzuki Y (2017) Ecological and genomic profiling of anaerobic methane-oxidizing archaea in a deep granitic environment. *ISME J*.

Lin Z, Sun X, Peckmann J, Lu Y, Xu L, Strauss H, Zhou H, Gong J, Lu H, Teichert Bma (2016) How sulfate-driven anaerobic oxidation of methane affects the sulfur isotopic composition of pyrite: A SIMS study from the South China Sea. *Chemical Geology*, 440, 26-41.

Niemann H, Elvert M (2008) Diagnostic lipid biomarker and stable carbon isotope signatures of microbial communities mediating the anaerobic oxidation of methane with sulphate. *Organic Geochemistry*, 39, 1668-1677.

Peckmann J, Thiel V (2004) Carbon cycling at ancient methane-seeps. *Chemical Geology*, 205, 443-467.

Sahlstedt E, Karhu Ja, Pitkänen P, Whitehouse M (2016) Biogenic processes in crystalline bedrock fractures indicated by carbon isotope signatures of secondary calcite. *Applied Geochemistry*, 67, 30-41.

Simkus Dn, Slater Gf, Lollar Bs, Wilkie K, Kieft Tl, Magnabosco C, Lau Mcy, Pullin Mj, Hendrickson Sb, Wommack Ke, Sakowski Eg, Heerden Ev, Kuloyo O, Linage B, Borgonie G, Onstott Tc (2016) Variations in microbial carbon sources and cycling in the deep continental subsurface. *Geochimica et Cosmochimica Acta*, 173, 264-283.

Ziegenbalg Sb, Birgel D, Hoffmann-Sell L, Pierre C, Rouchy Jm, Peckmann J (2012) Anaerobic oxidation of methane in hypersaline Messinian environments revealed by ¹³C-depleted molecular fossils. *Chemical Geology*, 292-293, 140-148.

Reviewer #2 (Remarks to the Author):

Congratulations on writing a very clear, well structured paper. The application of CAS to these materials is novel and has produced interesting data. The methods are appropriate, the results appear to be valid, however, I have two major points which you need to address.

1) The last sentence of the abstract reads "Therefore AOM may serve as an important, unrecognized methane sink that reduces methane emissions in salt dome settings perhaps associated with an extensive, deep subsurface biosphere." And you amplify this in the end of the Results and Conclusions section. While I understand your wish to make this very nice piece of science topical by relating it to greenhouse gas emissions, which might also make it more attractive to *Nature Geoscience*, it is not convincing. The next step, that you have not taken, would be to quantify this effect and show that it is significant. You could estimate bulk methane mineralisation from the amount of calcite but after that you would need to have estimates of methane seepage rates and be able to model the historical Evolution of the system in the amount of methane mineralised per unit time. My GUESS is that it would be very small but even the bulk amount might be an indicator of its significance.

2) You make a good case for the dominant reaction being oxidation of methane by sulphate to

produce calcite. You hypothesise that the sulphate reduction process is microbiological rather than thermochemical. This may be true but is not proven. Data on the isotopic fractionation of sulphur between sulphate and hydrogen sulphide as a result of thermochemical processes are relatively few and most are inferred from assumed thermochemical natural occurrences. It is probable that the range of fractionation factors would depend on many aspects of the local environmental conditions at the time of reaction. You would strengthen your case for your results being from the products microbial reactions if you were able to compare results from salt domes where the temperatures of sulphate reduction are known (for example, from fluid inclusion data from calcite and/or modelled burial temperatures) together with sulphide S-34 data. It would be much more convincing if you had data both from more likely thermochemical sulphate reduction and lower temperature microbiological processes. The real clincher would it be to relate the amount of methane oxidised to temperature. Ideally the amount would go down with increased temperature for a microbially process system (assuming that the microbial consortia have have a relatively normal temperature profile) but increase with temperature for thermochemical systems, like those in the Khuff reservoirs in the Middle East.

Reviewer #3 (Remarks to the Author):

As someone who has worked on salt dome cap rock genesis, I am glad to see some new interest in the subject and new data. I believe the paper could have broad appeal to scientists in other disciplines, as salt dome cap rocks are very interesting and their genesis are not all-that-well documented or studied in the past couple of decades.

I am not very familiar with the CAS procedure, and the results combined with $\delta^{13}\text{C}$ data are interesting. However, the concept that methane was the ultimate source of carbon in cap rock calcite is not new, nor are the interpreted (bio)geochemical reactions for cap rock genesis. And previous researchers couldn't rule out some contribution of the lighter, more easily biodegraded liquid hydrocarbons to the C-isotopic composition of the calcite. If the new data precludes that possibility, it would be worth pointing that out.

The proposal that this methane oxidation process in salt dome cap rocks could have significantly affected global methane gas concentrations in the atmosphere is new and perhaps innovative. But that needs to be quantified; how much methane was turned into calcite? Some mass balance calculations would prove helpful. My intuition is that this is not a major sink for methane (e.g. based on total mass of worldwide salt dome caprock calcite) but perhaps I am wrong.

A couple of specific comments:

1) Previous published research has shown that there are many "early" and "late" episodes of geochemical reactions, perhaps driven by episodic introduction of hydrocarbons into the caprock. So the process is not nearly as simplistic as this paper implies.

2) Several publications have documented or proposed that sedimentary formations waters (oil field brines) played an important role in cap rock formation. At the very least they appear to be the source of Fe, Pb, Zn, and Sr in cap rock minerals (perhaps with entrained hydrocarbons?). So if that is true, and such brines contain dissolved sulfate, then it is likely that such fluids can be trapped in fluid inclusions in the biogenic calcite. It appears that the CAS technique used here would liberate the fluid inclusion fluids to the solutions analyzed for their S-isotopic composition and that needs to be evaluated.

Calcite formation in salt dome cap rocks by microbial anaerobic oxidation of methane by Caesar et al.

Response to Review

Please find responses to the first round of review below. Included line numbers indicate changes to the text in the marked draft, where appropriate. Reviewer comments are provided in regular text, responses are provided in italics.

Reviewer #1 (Remarks to the Author):

This is a potentially interesting manuscript about anaerobic oxidation of methane (AOM) associated with salt domes (cap rocks). The authors claim that carbonates formed as cap rocks above these salt domes serve as unrecognized widespread sinks for hydrocarbons. The authors use mainly C and S isotopes in carbonate and carbon associated sulfate, respectively, to prove their point. Whereas the C isotope data indeed speaks in favor of an AOM process, the S isotope data and interpretations are more problematic. In its current form the manuscript does not convincingly prove microbial sulfate reduction or rule out thermochemical sulfate reduction. S isotope data are given and cited, but are not presented conclusively or in such detail that a full process understanding can be deduced. I have suggested a couple of avenues (below) that the authors could follow to prove their point more convincingly.

Additional sulfur isotope data from the reduced sulfur phases (elemental sulfur and sulfide mineral phases) have been collected. Clumped isotope data have also been collected. These new data in addition to the previously reported carbonate-hosted CAS and carbon isotope data indicate that microbial, sulfate-dependent AOM was a significant calcite producing reaction pathway (added and modified lines 21-25, 185-191, 202-234, 279-300, Figures 2, 4, 5, 6).

Furthermore, much more information about the system is needed (i.e. geological setting, textural documentation) to make the story comprehensive. As an example, the study is on carbonate minerals, but it is unclear whether it is a fully fossil system or feature ongoing processes. The system seems to have evolved over time. Are there any age constraints on these different phases and/or on the process as a whole?

Additional discussion of paragenetic relationships has been added (lines 202-234), including the new Figure 4. Geochemical data in Figures 5 and 6 have been distinguished based on phase and paragenetic relationship.

Aside from these relative time indicators, there are no absolute age constraints on salt dome cap rock phases. As a result, we are left to rely on spatial mineralogical relationships to establish precipitation timing.

It is also not clear why only methane oxidation is able to form carbonate cap rocks when other processes can produce bicarbonate as well and other carbon sources are indicated in the isotope data of the carbonates (not all carbonate-C is ^{13}C -depleted).

The potential for other reaction pathways to produce calcite in cap rock systems is not specifically rejected. Indeed, as the reviewer states, the range in $\delta^{13}\text{C}$ data suggests that carbon is derived from multiple sources in some instances (as discussed in lines 243-245, 325-332). The fact that most data, including at least one data point from each locality, fall below the liquid hydrocarbon minimum (Figure 5a) supports the widespread occurrence of methane oxidation. In fact, data that fall outside the AOM window in Figure 5a may still reflect partial incorporation of methane oxidation sourced carbon (lines 245-248).

Substantial revision and addition of analytical data (see bullets below), sample descriptions (preferentially the relation between pyrite and calcite) and geological setting are needed before it can be recommended for publication.

Additional analytical data, relevant discussion and paragenetic relationship identification have been added to the revised manuscript (refer to other comment responses for specific line modifications).

It is recommended to discuss and compare the findings in the context of other recently described non-marine methane sources and sinks (such as those described in various works by Etiope and colleagues (Etiope & Sherwood Lollar, 2013 ; Etiope, 2009 ; Etiope & Klusman, 2002) and the similarly operating carbonate mineral sink for methane recently discovered in the upper continental crust (Drake et al., 2017 ; Drake et al., 2015 ; Sahlstedt et al., 2016), which also have been discussed from microbiological perspectives (Ino et al., 2017 ; Bomberg et al., 2015 ; Simkus et al., 2016) and not only comparing the data with marine findings.

Citations of other AOM settings have been added to the manuscript (lines 347-352). Discussion about how the salt dome environment is distinguishable from the other settings is also provided (lines 352-358).

To conclude, the manuscript is interesting and the recognition of a widespread methane sink in salt dome cap rocks has wide implications, but the methods used should be expanded and the data more carefully presented/interpreted in order to be in line with the high standard of Nature Communications. The journal has quite generous word limit allowing the authors to add more details.

Specific remarks:

1. As presented, the S-isotope data seems to be used as a diagnostic tool for AOM. For instance, there is a field in fig.4 “AOM Field”, but strictly speaking there are no specific S-isotope values that are diagnostic for AOM. Yes, S-isotopes can be used to prove microbial sulfate reduction (MSR) but not independently to prove AOM. This is also indicated in the text at lines 246-249 describing similar range of S-isotope values from diapirs without methane oxidation.

The sulfur isotope data alone was never meant to indicate AOM diagnostically. Rather, the integrated use of $\delta^{34}S_{CAS}$ and $\delta^{13}C_{carb}$ is used to define a data window reflective of sulfate-dependent AOM (the field is expressed in both $\delta^{13}C$ and $\delta^{34}S$ space in Figure 5a). To help clarify this, the AOM Field labeling has been changed.

2. The heavy S isotope values are used to prove MSR in closed system. However, there are a wide range of values that needs to be discussed to understand the whole system. Heavy d34S values can be due to closed system Rayleigh fractionation but then what about MSR that is not undergoing closed system fractionation? This cannot be distinguished using sulfate d34S only but might still be occurring.

We agree with the reviewer that open system sulfate reduction may be occurring. However, the fact that nearly all samples exhibit $\delta^{34}S_{CAS} > +16\text{‰}$ indicates the predominance of closed-system sulfate reduction. Therefore detailed discussion of open system sulfate reduction is not warranted.

A much more straightforward way to gain knowledge of MSR is to study the product of MSR (i.e. sulfide minerals) instead of the residual sulfate alone. By using micro-scale analysis in pyrite (e.g. SIMS) it is possible to distinguish the evolution of a closed-system within the setting (see e.g. Drake et al., 2015; Lin et al., 2016). If there was abundant MSR (related to AOM) there should also be pyrite present. The authors use d34S in pyrite from cited data but i) do not present it comprehensively (instead presenting it grouped with other S-minerals) and ii) state that it is not certain whether pyrite and calcite are co-genetic. So why used it at all in that case? It is recommended instead to do a thorough check in the samples if pyrite/calcite paragenesis can be distinguished and then analyze the pyrite for d34S (micro-scale analysis preferred).

We have now completed a thorough assessment of the paragenetic relationships between calcite and the reduced sulfur phases (elemental sulfur and sulfide) in these samples (lines 202-229, Figure 4). In most instances, these phases do not appear to be cogenetic with calcite. We feel that this difference in precipitation timing is the primary reason why using carbonate-hosted proxies ($\delta^{13}C_{carb}$ and $\delta^{34}S_{CAS}$) is a more straightforward way to assess how S-C cycling relates to calcite precipitation (as stated in lines 229-234).

3. d34S in CAS is heavy (at least partly, based on the presented data). This means that in order to have mass balance there must be significant amounts of isotopically light sulfide (or some other S-phase) precipitated in the system. The discussion should be expanded to include this mass balance problem.

We now present sulfur isotope data from reduced sulfur phases and demonstrate that they exhibit $\delta^{34}S$ values that are lower than minor sulfate that occurs in the Louann Salt. It should be noted that the product ^{34}S -depleted sulfide need not be precipitated as a mineral phase in order to promote Rayleigh-type sulfate ^{34}S -enrichment in residual sulfate. Rather, this simply requires isolation of the product sulfide, whether it be through mineral precipitation or a lack of reoxidation to sulfate.

4. The span in d34S values can theoretically be due to thermochemical sulfate reduction (TSR). This is ruled out by the authors due to that pyrite has very low d34S values (i.e. too large fractionation for abiotic processes). But there were, again, difficulties in assessing whether pyrite (cited data) and calcite were co-genetic so clearly the temporal pyrite-calcite relation needs to be strengthened and/or additional proof for MSR should be added. What about fluid inclusions in the calcites? That could prove that the calcites are low-T and rule out TSR.

We have included new clumped isotope data to reconcile this issue. These new data indicate temperatures that are too low to accommodate thermochemical sulfate reduction and thus support the original microbial interpretations. See lines 279-300, Figure 6.

5. It is not clear if methane was oxidized by microbes or result of abiotic processes. Fluid inclusion homogenization temperatures could support this interpretation as well, if the temperatures are low microbial methanotrophy is likely.

See clumped isotope comment above.

In addition, it is well known from fossil AOM-MSR systems that organic remains are present within authigenic carbonates. There are several proxies that can be used (biomarkers, like specific fatty acids as well as isotope signatures of different compounds, see e.g. Niemann & Elvert, 2008; Elvert et al., 2003; Ziegenbalg et al., 2012)). The manuscript would be significantly strengthened by inclusion fluid inclusion analysis and/or analyses of any remains of microbial communities.

Agreed. Prior to the original manuscript submittal, we attempted organic carbon content and isotope analyses on these samples. Unfortunately the organic yields were too low to produce accurate data.

6. Statement that AOM is needed for carbonate cap rock formation. This is not fully supported by the data, because 1) the $\delta^{13}\text{C}$ values of the carbonate (cited and new data) can be as heavy as -2 permil which certainly is not related to AOM, so other sources of alkalinity exist.

We fully acknowledge that there may be additional sources of carbon for calcite precipitation (lines 243-248, 325-332). The common occurrence of $\delta^{13}\text{C}_{\text{carb}}$ values below -28‰ indicates that methane oxidation is a widespread process that leads to carbonate precipitation. In reality even a sample with a $\delta^{13}\text{C}_{\text{carb}}$ value of -2‰ could have received some carbon from methane oxidation, albeit not that much.

2) there is evidently MSR from organic matter oxidation in salt dome/cap rocks as cited in the manuscript. This process could lead to HCO_3^- formation and hence in an increased alkalinity that would initiate carbonate precipitation.

We agree that organic matter-related MSR could lead to the precipitation of authigenic carbonate in principle. However, we do not find direct evidence for this in GMB cap rock. We also stress that other systems where organic matter-related MSR is occurring do not exhibit cap rock calcite (335-343).

3) Not all sites plot in the “AOM field” of Fig. 4. 4) The assumption that all calcite with values below -28 permil is shaky. There is fractionation to the lighter side when organic matter and oil are degraded by microbes (microbes use the lighter isotope preferentially), so the bicarbonate produced in this way can be lighter than -28 permil. There are clearly

some borderline cases in that $\delta^{13}\text{C}$ range in fig 4. More discussion and more careful presentation on this is needed.

We acknowledge that there can be fractionations associated with organic matter and liquid hydrocarbon oxidation. We now cite studies that explore this (lines 238-243). We argue that in most instances these fractionations are relatively small and are unlikely to discount the predominance of methane-sourced carbon for carbonate precipitation.

7. Addition of geochronological constraints would be useful.

See above.

Minor suggestions:

1. The process behind the abundant precipitation of elemental S should be described.

The sulfur formation mechanism is mentioned (lines 125-126). We feel a detailed discussion of elemental sulfur formation processes is outside the scope of this research.

2. Describe more comprehensively in the introduction how the AOM+SR process works in general (based on findings e.g. at SMTZ in marine sediments and in continental crust).

We feel that the relevant reaction pathways are sufficiently presented and discussed. Detailed description of different environments where these reactions occur would detract from the manuscript. The reader is able to refer to the included references for additional information.

3. Fig. 2, mark the mineral phases in the figure.

The updated Figure 2 and new Figure 4 include marked mineral phases.

4. Line 151. Fractionation of $\delta^{13}\text{C}$ during AOM should also be taken into account, as well as mixing of C from AOM with other carbon sources, which taken together as a rule make $\delta^{13}\text{C}$ composition of authigenic carbonate different from the source methane (usually heavier methane, see Peckmann & Thiel, 2004).

The proposed negative carbon isotope fractionation upon methane oxidation reported in Peckmann and Thiel (2004) after work by Hovland et al. (1987) does not appear to be a widespread phenomena. In fact, most of the discussion in Peckmann and Thiel (2004) emphasizes that carbonates generally incorporate similar or slightly ^{13}C -enriched values compared to the host methane. They state:

“Where both carbonate minerals and methane gas have been analysed at modern seeps, an enrichment of ^{13}C in the carbonate relative to the gas was usually found. Comparing the most negative $\delta^{13}\text{C}_{\text{carbonate}}$ values of each deposit with the respective $\delta^{13}\text{C}_{\text{methane}}$ values, it appears that most of the $C_{\text{carbonate}}$ derived from methane, but other carbon sources are still significant (Table 1). Only methane from the North Sea yielded exclusively higher $\delta^{13}\text{C}$ values than associated seep carbonates (Hovland et al., 1987)”.

This generalized relationship can also be seen in their Table 1.

Ultimately, the very light $\delta^{13}\text{C}_{\text{carb}}$ values reported here are frequently attributed to methane oxidation-sourced carbon. In instances where values exceed the lower limit for hydrocarbon $\delta^{13}\text{C}$, we fully acknowledge that additional carbon sources may be important.

5. Lines 164-172, describe the S-isotope data in greater detail by writing out values for different mineral phases and giving numerical information about what is meant by “large isotope fractionation”.

The new Figure 5 includes sulfur isotope data from all relevant phases. Discussion of what is considered a large isotope fractionation is provided in lines 295-298.

Reviewer #2 (Remarks to the Author):

Congratulations on writing a very clear, well structured paper. The application of CAS to these materials is novel and has produced interesting data. The methods are appropriate, the results appear to be valid, however, I have two major points which you need to address.

1) The last sentence of the abstract reads "Therefore AOM may serve as an important, unrecognized methane sink that reduces methane emissions in salt dome settings perhaps associated with an extensive, deep subsurface biosphere." And you amplify this in the end of the Results and Conclusions section. While I understand your wish to make this very nice piece of science topical by relating it to greenhouse gas emissions, which might also make it more attractive to Nature Geoscience, it is not convincing. The next step, that you have not taken, would be to quantify this effect and show that it is significant. You could estimate bulk methane mineralisation from the amount of calcite but after that you would need to have estimates of methane seepage rates and be able to model the historical Evolution of the system in the amount of methane mineralised per unit time. My GUESS is that it would be very small but even the bulk amount might be an indicator of its significance.

We have added discussion concerning quantification of the amount and rate of this methane sink. Although constraints are admittedly loose (as we directly state in lines

369-396), it appears that the reviewer's guess is correct about slow rates (see lines 387-399, Supplemental Information). We redirect the significance of methane oxidation as it relates the shear abundance of mineral produced in the salt dome environment of the GMB (lines 396-399).

2) You make a good case for the dominant reaction being oxidation of methane by sulphate to produce calcite. You hypothesise that the sulphate reduction process is microbiological rather than thermochemical. This may be true but is not proven. Data on the isotopic fractionation of sulphur between sulphate and hydrogen sulphide as a result of thermochemical processes are relatively few and most are inferred from assumed thermochemical natural occurrences. It is probable that the range of fractionation factors would depend on many aspects of the local environmental conditions at the time of reaction. You would strengthen your case for your results being from the products microbial reactions if you were able to compare results from salt domes where the temperatures of sulphate reduction are known (for example, from fluid inclusion data from calcite and/or modelled burial temperatures) together with sulphide S-34 data. It would be much more convincing if you had data both from more likely thermochemical sulphate reduction and lower temperature microbiological processes. The real clincher would it be to relate the amount of methane oxidised to temperature. Ideally the amount would go down with increased temperature for a microbially process system (assuming that the microbial consortia have have a relatively normal temperature profile) but increase with temperature for thermochemical systems, like those in the Khuff reservoirs in the Middle East.

In order to distinguished between thermochemical and microbial processes we have added clumped isotope data. Relatively low temperature reconstructions (lower than thermochemical sulfate reduction ranges) based on this new data support the previous inference of a microbially dominated system. See lines 279-300, Figure 6.

Reviewer #3 (Remarks to the Author):

As someone who has worked on salt dome cap rock genesis, I am glad to see some new interest in the subject and new data. I believe the paper could have broad appeal to scientists in other disciplines, as salt dome cap rocks are very interesting and their genesis are not all-that-well documented or studied in the past couple of decades.

I am not very familiar with the CAS procedure, and the results combined with $\delta^{13}\text{C}$ data are interesting. However, the concept that methane was the ultimate source of carbon in cap rock calcite is not new, nor are the interpreted (bio)geochemical reactions for cap rock genesis. And previous researchers couldn't rule out some contribution of the lighter, more easily biodegraded liquid hydrocarbons to the C-isotopic composition of the calcite.

If the new data precludes that possibility, it would be worth pointing that out.

The new CAS data allows for assessment of coupled sulfur and carbon reactions, rather than providing new insights into carbon sources specifically. When the data are considered as a whole, it seems likely that at least some calcite receives carbon from multiple sources (lines 235-248, 325-334). We rely on the predominance of $\delta^{13}C_{carb}$ values lower than the reported light end-member of GMB petroleum (including at least one sample from each location) to assert that methane is likely a widespread carbon source (Figure 5a).

The proposal that this methane oxidation process in salt dome cap rocks could have significantly affected global methane gas concentrations in the atmosphere is new and perhaps innovative. But that needs to be quantified; how much methane was turned into calcite? Some mass balance calculations would prove helpful. My intuition is that this is not a major sink for methane (e.g. based on total mass of worldwide salt dome caprock calcite) but perhaps I am wrong.

As with the previous reviewer, this reviewer's assertion about methane oxidation rate is probably correct (lines 369-396, Supplementary Information). Again, we redirect the significance of methane oxidation as it relates the shear abundance of mineral produced in the salt dome environment of the GMB (lines 396-399).

A couple of specific comments:

1) Previous published research has shown that there are many “early” and “late” episodes of geochemical reactions, perhaps driven by episodic introduction of hydrocarbons into the caprock. So the process is not nearly as simplistic as this paper implies.

We have added additional paragenesis data and discussion in order to account for variable mineral formation timing. Figures have also been modified to include phase paragenetic relationships. See lines 202-229, Figures 4, 5, 6)

2) Several publications have documented or proposed that sedimentary formations waters (oil field brines) played an important role in cap rock formation. At the very least they appear to be the source of Fe, Pb, Zn, and Sr in cap rock minerals (perhaps with entrained hydrocarbons?). So if that is true, and such brines contain dissolved sulfate, then it is likely that such fluids can be trapped in fluid inclusions in the biogenic calcite. It appears that the CAS technique used here would liberate the fluid inclusion fluids to the solutions analyzed for their S-isotopic composition and that needs to be evaluated.

The CAS extraction procedure is developed in part to remove non-lattice bound sulfate prior to analysis. This includes phases that are incorporated as fluid inclusions. We are confident that potential fluid inclusion sulfate is not contributing to the reported $\delta^{34}S_{CAS}$ data.

Reviewers' comments:

Reviewer #1 (Remarks to the Author):

I reviewed a previous version of this manuscript (reviewer#1). The authors have made a serious effort to improve the manuscript, for instance by adding clumped isotope data that strengthen the manuscript.

The paper is well structured and easy to follow. It is perhaps a bit general and simplified, but I do not have a problem with that. I have some small remarks listed below which I hope the authors consider, and after that I think this manuscript can be accepted.

Small remarks:

Line 121-122: It is puzzling that this process "hydrocarbon species react with sulfate" is not being more detailed in the previous literature. It would be good if the authors here could present a couple of lines with a little bit more in depth presentation of previous models for the carbonate cap rock precipitation (surely there must be some more details?). All the cited literature in this section is 30-50 years old. This is not a setting that I usually work with but I assume that there are some more detailed/updated models, at least from other analogue areas (such as those listed in the introduction?).

Line 195-196: Previously reported data...reference needed here.

Line 200: "34S enrichments consistent with sulfate reduction", yes, but a bit oversimplified, since this only applies if this is residual sulfate following Rayleigh fractionation reservoir effects, or at least that the produced sulfide is isolated from sulfate. Please be a bit more more specific here.

Line 211: "Radiating crystals mimic calcite..." are you sure that this was not primary aragonite? The radiating crystals are more typical for aragonite. Please provide some evidence (direct or circumstantial) that it was calcite and not aragonite.

Line 387-399: The estimation of methane oxidation is of course very general as there are so many unknowns, uncertainties and extrapolations of these, but still of importance. It could be important to emphasize though that 1) calcite precipitation is probably to some degree episodic in nature, with periods of more precipitation when there are tectonic events or similar that induce fluid mixing, gas migration and exposure of fresh sulfates to alter, 2) that over a time frame of 165 Ma, theoretically, periods also of dissolution of calcite can be expected(?), this would underestimate the methane sink estimated from AOM-related calcite abundance.

Suppl info:

Fig: label panels, a, b, c and describe the difference between them more carefully in the caption. It is difficult to follow the reasoning in the caption without reference to sub-panel labeling.

Reviewer #3 (Remarks to the Author):

I reviewed an earlier version of the paper and found that in the latest version the authors adequately addressed my concerns about how much methane was "kept" from the atmosphere. As I suspected, that would probably not be a huge amount by global standards. However, I have little doubt that bacterial sulfate-reduction (BSR) was the principal process involved in methane oxidation, which is consistent with the C, S-isotope data presented. I recommend that the paper be published with some minor revisions.

I do have a few more comments the authors might want to consider in preparing their final version of the manuscript.

1) Mineral paragenesis: Calcite cap rock is not a monolithic entity as it formed by the progressive dissolution of underlying anhydrite and results in a banded texture reflecting what has been called

“inverted stratigraphy” (oldest layers on the TOP) as has been well documented by coauthor Kyle and his colleagues. Thus paragenetic relationships are much more complex than can be illustrated by Figure 4. For example, calcite veinlets that cross-cut other minerals and thus are younger, may have their “roots” in coarser-grained, euhedral calcite forming just below. Further, because open spaces abound in the calcite cap rock, minerals that fill the voids can be younger than the minerals from the walls of the voids. Overall, it looks to me like sulfide minerals formed more or less contemporaneous with the calcite in any specific layer of the calcite cap rock. Indeed, the sulfide minerals are what given some of the bands their dark color. In contrast, SULFATE minerals (barite, celestine) always appear to post-date nearby calcite and sulfides. That observation is entirely compatible with a Rayleigh fractionation trend caused by the apparent bacterial sulfate reduction for the sulfur isotopes (consistent with the author’s data). I think it is important to give some more details about the paragenetic details of the calcite cap rock formation as it is related to the formation of this inverted stratigraphy.

2) Temperature of calcite cap rock formation. The authors are probably correct in their conclusion that the temperature of formation of the calcite cap rock is too low for in situ thermochemical sulfate reduction. In addition to their interpreted isotope data, they describe the ubiquitous presence of single-phase fluid inclusions in calcite as additional evidence of a low temperature of formation. Saunders and Swann (1994) presented some quantitative homogenization temperatures from relatively large fluid inclusions in barite, from Hazlehurst salt dome, MS and found a maximum temperature of ~55oC for 2-phase (L+V). However, they interpreted the formation of the 2-phase fluid inclusions to be the result of an artifact of sample preparation. In short, polishing of the samples led to stretching of barite causing the formation of the 2-phase fluid inclusions. However, when such stretching occurs, it yields anomalously HIGH homogenization temperatures, and thus they concluded cap rock minerals formed at < ~50oC. So perhaps that study has some bearing on the temperature of formation of the calcite cap rocks studied in this paper.

3) Sulfur isotopes of sulfide minerals in calcite cap rocks. They typically range from ~0 to +10 per mil for $\delta^{34}\text{S}$, a range which is seemingly “too heavy” to have been the result of bacterial sulfate reduction in comparison to the “normal” range of values for sedimentary sulfides (e.g. pyrite) formed by BSR. Saunders and Swann (1994) proposed that isotopically heavy sulfur left over from a previous Rayleigh fraction cycle (from an earlier calcite layer formation) mixed with new sulfate derived from the cap rock anhydrite and incoming formation waters (brines) to produce a starting $\delta^{34}\text{S}$ of dissolved sulfate perhaps 10-15 per mil heavier than cap-rock anhydrite, and thus a new cycle of BSR would yield sulfide minerals in the 0-10 per mil range.

Reference:

Saunders, J.A., and Swann, C.T., 1994, Mineralogy and geochemistry of cap rock Zn-Pb-Sr-Ba mineralization at Hazlehurst salt dome, Mississippi: *Economic Geology*, v. 89, p. 381-390.

Review by J.A. Saunders

Carbonate formation in salt dome cap rocks by microbial anaerobic oxidation of methane by Caesar et al.

Response to Second Round of Review

Please find responses to the second round of review below. Included line numbers indicate changes to the text in the revised draft, where appropriate. Reviewer comments are provided in regular text, responses are provided in italics.

Reviewer #1 (Remarks to the Author):

I reviewed a previous version of this manuscript (reviewer#1). The authors have made a serious effort to improve the manuscript, for instance by adding clumped isotope data that strengthen the manuscript.

The paper is well structured and easy to follow. It is perhaps a bit general and simplified, but I do not have a problem with that. I have some small remarks listed below which I hope the authors consider, and after that I think this manuscript can be accepted.

Small remarks:

Line 121-122: It is puzzling that this process “hydrocarbon species react with sulfate” is not being more detailed in the previous literature. It would be good if the authors here could present a couple of lines with a little bit more in depth presentation of previous models for the carbonate cap rock precipitation (surely there must be some more details?). All the cited literature in this section is 30-50 years old. This is not a setting that I usually work with but I assume that there are some more detailed/updated models, at least from other analogue areas (such as those listed in the introduction?).

Gulf Coast cap rock research has not been very active in the past two decades. This is particularly true about research devoted to specification of carbonate mineralization pathways. We surmise that this may be related to delayed development of new proxy tools (such as the carbonate-associated sulfate and clumped isotope approaches used here) that would allow confounding issues to be addressed. Ultimately there are no “recent” articles that address reaction specifics and we are left with a relatively old literature base.

Line 195-196: Previously reported data...reference needed here.

References added to line 200.

Line 200: “³⁴S enrichments consistent with sulfate reduction”, yes, but a bit oversimplified, since this only applies if this is residual sulfate following Rayleigh fractionation reservoir effects, or at least that the produced sulfide is isolated from sulfate. Please be a bit more more specific here.

Line 206 has been modified.

Line 211: “Radiating crystals mimic calcite...” are you sure that this was not primary aragonite? The radiating crystals are more typical for aragonite. Please provide some evidence (direct or circumstantial) that it was calcite and not aragonite.

References to calcite throughout the manuscript have been changed to carbonate to be more generic. In this example, we agree with the reviewer that the radiating crystal habit likely reflects aragonite. This is now specifically stated in lines 225-227.

Line 387-399: The estimation of methane oxidation is of course very general as there are so many unknowns, uncertainties and extrapolations of these, but still of importance. It could be important to emphasize though that 1) calcite precipitation is probably to some degree episodic in nature, with periods of more precipitation when there are tectonic events or similar that induce fluid mixing, gas migration and exposure of fresh sulfates to alter, 2) that over a time frame of 165 Ma, theoretically, periods also of dissolution of calcite can be expected(?), this would underestimate the methane sink estimated from AOM-related calcite abundance.

All good and valid points. We have added lines 410-414.

Suppl info:

Fig: label panels, a, b, c and describe the difference between them more carefully in the caption. It is difficult to follow the reasoning in the caption without reference to sub-panel labeling.

Panel labels and additional discussion have been added to the supplemental figure and its caption.

Reviewer #3 (Remarks to the Author):

I reviewed an earlier version of the paper and found that in the latest version the authors adequately addressed my concerns about how much methane was “kept” from the atmosphere. As I suspected, that would probably not be a huge amount by global standards. However, I have little doubt that bacterial sulfate-reduction (BSR) was the principal process involved in methane oxidation, which is consistent with the C, S-isotope data presented. I recommend that the paper be published with some minor revisions.

I do have a few more comments the authors might want to consider in preparing their final version of the manuscript.

1) Mineral paragenesis: Calcite cap rock is not a monolithic entity as it formed by the progressive dissolution of underlying anhydrite and results in a banded texture reflecting what has been called “inverted stratigraphy” (oldest layers on the TOP) as has been well documented by coauthor Kyle and his colleagues. Thus paragenetic relationships are much more complex than can be illustrated by Figure 4. For example, calcite veinlets that cross-cut other minerals and thus are younger, may have their “roots” in coarser-grained, euhedral calcite forming just below. Further, because open spaces abound in the calcite cap rock, minerals that fill the voids can be younger than the minerals from the walls of the voids. Overall, it looks to me like sulfide minerals formed more or less contemporaneous with the calcite in any specific layer of the calcite cap rock. Indeed, the sulfide minerals are what given some of the bands their dark color. In contrast, SULFATE minerals (barite, celestine) always appear to post-date nearby calcite and sulfides. That observation is entirely compatible with a Rayleigh fractionation trend caused by the apparent bacterial sulfate reduction for the sulfur isotopes (consistent with the author’s data). I think it is important to give some more details about the paragenetic details of the calcite cap rock formation as it is related to the formation of this inverted stratigraphy.

This is valid. We have added lines 128-130, 145-146 and 239-246 to address these points.

2) Temperature of calcite cap rock formation. The authors are probably correct in their conclusion that the temperature of formation of the calcite cap rock is too low for in situ thermochemical sulfate reduction. In addition to their interpreted isotope data, they describe the ubiquitous presence of single-phase fluid inclusions in calcite as additional evidence of a low temperature of formation. Saunders and Swann (1994) presented some quantitative homogenization temperatures from relatively large fluid inclusions in barite, from Hazlehurst salt dome, MS and found a maximum temperature of ~55oC for 2-phase (L+V). However, they interpreted the formation of the 2-phase fluid inclusions to be the result of an artifact of sample preparation. In short, polishing of the samples led to stretching of barite causing the formation of the 2-phase fluid inclusions. However, when such stretching occurs, it yields anomalously HIGH homogenization temperatures, and thus they concluded cap rock minerals formed at <~50oC. So perhaps that study has

some bearing on the temperature of formation of the calcite cap rocks studied in this paper.

These findings and those within the current manuscript support the idea that multiple (rather than just carbonate) phases exhibit temperature proxy data that indicate relatively low temperature conditions. We have added lines 312-318 to stress this.

3) Sulfur isotopes of sulfide minerals in calcite cap rocks. They typically range from ~0 to +10 per mil for $\delta^{34}\text{S}$, a range which is seemingly “too heavy” to have been the result of bacterial sulfate reduction in comparison to the “normal” range of values for sedimentary sulfides (e.g. pyrite) formed by BSR. Saunders and Swann (1994) proposed that isotopically heavy sulfur left over from a previous Rayleigh fraction cycle (from an earlier calcite layer formation) mixed with new sulfate derived from the cap rock anhydrite and incoming formation waters (brines) to produce a starting $\delta^{34}\text{S}$ of dissolved sulfate perhaps 10-15 per mil heavier than cap-rock anhydrite, and thus a new cycle of BSR would yield sulfide minerals in the 0-10 per mil range.

We have added lines 275-280. These discuss the sulfur isotope evolution of sulfide phases in more detail and the relation to sulfide $\delta^{34}\text{S}$ values near 0‰.

REVIEWERS' COMMENTS:

Reviewer #1 (Remarks to the Author):

The authors have done a good job revising the manuscript according to my remarks. To me, this manuscript can now be accepted for publication.

//

Henrik Drake

Reviewer #3 (Remarks to the Author):

Good job.....I believe the authors have adequately responded to the reviewers concerns, and I recommend that the paper be published now with just an editorial review of grammar an spelling.

Jim Saunders
Auburn, AL USA